# Spatial Transformer Networks

**Max Jaderberg**     **Karen Simonyan**     **Andrew Zisserman**     **Koray Kavukcuoglu**

Google DeepMind, London, UK
{jaderberg,simonyan,zisserman,korayk}@google.com

## Abstract

Convolutional Neural Networks define an exceptionally powerful class of models, but are still limited by the lack of ability to be spatially invariant to the input data in a computationally and parameter efficient manner. In this work we introduce a new learnable module, the *Spatial Transformer*, which explicitly allows the spatial manipulation of data within the network. This differentiable module can be inserted into existing convolutional architectures, giving neural networks the ability to actively spatially transform feature maps, conditional on the feature map itself, without any extra training supervision or modification to the optimisation process. We show that the use of spatial transformers results in models which learn invariance to translation, scale, rotation and more generic warping, resulting in state-of-the-art performance on several benchmarks, and for a number of classes of transformations.

## 1   Introduction

Over recent years, the landscape of computer vision has been drastically altered and pushed forward through the adoption of a fast, scalable, end-to-end learning framework, the Convolutional Neural Network (CNN) [18]. Though not a recent invention, we now see a cornucopia of CNN-based models achieving state-of-the-art results in classification, localisation, semantic segmentation, and action recognition tasks, amongst others.

A desirable property of a system which is able to reason about images is to disentangle object pose and part deformation from texture and shape. The introduction of local max-pooling layers in CNNs has helped to satisfy this property by allowing a network to be somewhat spatially invariant to the position of features. However, due to the typically small spatial support for max-pooling (*e.g.* $2 \times 2$ pixels) this spatial invariance is only realised over a deep hierarchy of max-pooling and convolutions, and the intermediate feature maps (convolutional layer activations) in a CNN are not actually invariant to large transformations of the input data [5, 19]. This limitation of CNNs is due to having only a limited, pre-defined pooling mechanism for dealing with variations in the spatial arrangement of data.

In this work we introduce the *Spatial Transformer* module, that can be included into a standard neural network architecture to provide spatial transformation capabilities. The action of the spatial transformer is conditioned on individual data samples, with the appropriate behaviour learnt during training for the task in question (without extra supervision). Unlike pooling layers, where the receptive fields are fixed and local, the spatial transformer module is a dynamic mechanism that can actively spatially transform an image (or a feature map) by producing an appropriate transformation for each input sample. The transformation is then performed on the entire feature map (non-locally) and can include scaling, cropping, rotations, as well as non-rigid deformations. This allows networks which include spatial transformers to not only select regions of an image that are most relevant (attention), but also to transform those regions to a canonical, expected pose to simplify inference in the subsequent layers. Notably, spatial transformers can be trained with standard back-propagation, allowing for end-to-end training of the models they are injected in.

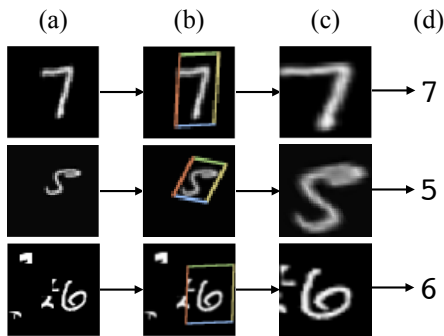

(a)     (b)     (c)     (d)

Figure 1: The result of using a spatial transformer as the first layer of a fully-connected network trained for distorted MNIST digit classification. (a) The input to the spatial transformer network is an image of an MNIST digit that is distorted with random translation, scale, rotation, and clutter. (b) The localisation network of the spatial transformer predicts a transformation to apply to the input image. (c) The output of the spatial transformer, after applying the transformation. (d) The classification prediction produced by the subsequent fully-connected network on the output of the spatial transformer. The spatial transformer network (a CNN including a spatial transformer module) is trained end-to-end with only class labels – no knowledge of the groundtruth transformations is given to the system.

Spatial transformers can be incorporated into CNNs to benefit multifarious tasks, for example: (i) *image classification:* suppose a CNN is trained to perform multi-way classification of images according to whether they contain a particular digit – where the position and size of the digit may vary significantly with each sample (and are uncorrelated with the class); a spatial transformer that crops out and scale-normalizes the appropriate region can simplify the subsequent classification task, and lead to superior classification performance, see Fig. 1; (ii) *co-localisation:* given a set of images containing different instances of the same (but unknown) class, a spatial transformer can be used to localise them in each image; (iii) *spatial attention:* a spatial transformer can be used for tasks requiring an attention mechanism, such as in [11, 29], but is more flexible and can be trained purely with backpropagation without reinforcement learning. A key benefit of using attention is that transformed (and so attended), lower resolution inputs can be used in favour of higher resolution raw inputs, resulting in increased computational efficiency.

The rest of the paper is organised as follows: Sect. 2 discusses some work related to our own, we introduce the formulation and implementation of the spatial transformer in Sect. 3, and finally give the results of experiments in Sect. 4. Additional experiments and implementation details are given in the supplementary material or can be found in the arXiv version.

## 2 Related Work

In this section we discuss the prior work related to the paper, covering the central ideas of modelling transformations with neural networks [12, 13, 27], learning and analysing transformation-invariant representations [3, 5, 8, 17, 19, 25], as well as attention and detection mechanisms for feature selection [1, 6, 9, 11, 23].

Early work by Hinton [12] looked at assigning canonical frames of reference to object parts, a theme which recurred in [13] where 2D affine transformations were modeled to create a generative model composed of transformed parts. The targets of the generative training scheme are the transformed input images, with the transformations between input images and targets given as an additional input to the network. The result is a generative model which can learn to generate transformed images of objects by composing parts. The notion of a composition of transformed parts is taken further by Tieleman [27], where learnt parts are explicitly affine-transformed, with the transform predicted by the network. Such generative capsule models are able to learn discriminative features for classification from transformation supervision.

The invariance and equivariance of CNN representations to input image transformations are studied in [19] by estimating the linear relationships between representations of the original and transformed images. Cohen & Welling [5] analyse this behaviour in relation to symmetry groups, which is also exploited in the architecture proposed by Gens & Domingos [8], resulting in feature maps that are more invariant to symmetry groups. Other attempts to design transformation invariant representations are scattering networks [3], and CNNs that construct filter banks of transformed filters [17, 25]. Stollenga *et al.* [26] use a policy based on a network's activations to gate the responses of the network's filters for a subsequent forward pass of the same image and so can allow attention to specific features. In this work, we aim to achieve invariant representations by manipulating the data rather than the feature extractors, something that was done for clustering in [7].

Neural networks with selective attention manipulate the data by taking crops, and so are able to learn translation invariance. Work such as [1, 23] are trained with reinforcement learning to avoid the

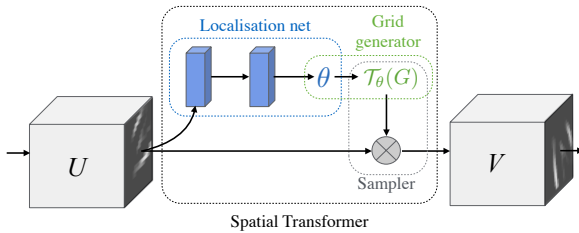

Figure 2: The architecture of a spatial transformer module. The input feature map $U$ is passed to a localisation network which regresses the transformation parameters $\theta$. The regular spatial grid $G$ over $V$ is transformed to the sampling grid $\mathcal{T}_\theta(G)$, which is applied to $U$ as described in Sect. 3.3, producing the warped output feature map $V$. The combination of the localisation network and sampling mechanism defines a spatial transformer.

need for a differentiable attention mechanism, while [11] use a differentiable attention mechansim by utilising Gaussian kernels in a generative model. The work by Girshick *et al.* [9] uses a region proposal algorithm as a form of attention, and [6] show that it is possible to regress salient regions with a CNN. The framework we present in this paper can be seen as a generalisation of differentiable attention to any spatial transformation.

# 3   Spatial Transformers

In this section we describe the formulation of a *spatial transformer*. This is a differentiable module which applies a spatial transformation to a feature map during a single forward pass, where the transformation is conditioned on the particular input, producing a single output feature map. For multi-channel inputs, the same warping is applied to each channel. For simplicity, in this section we consider single transforms and single outputs per transformer, however we can generalise to multiple transformations, as shown in experiments.

The spatial transformer mechanism is split into three parts, shown in Fig. 2. In order of computation, first a *localisation network* (Sect. 3.1) takes the input feature map, and through a number of hidden layers outputs the parameters of the spatial transformation that should be applied to the feature map – this gives a transformation conditional on the input. Then, the predicted transformation parameters are used to create a sampling grid, which is a set of points where the input map should be sampled to produce the transformed output. This is done by the *grid generator*, described in Sect. 3.2. Finally, the feature map and the sampling grid are taken as inputs to the *sampler*, producing the output map sampled from the input at the grid points (Sect. 3.3).

The combination of these three components forms a spatial transformer and will now be described in more detail in the following sections.

## 3.1   Localisation Network

The localisation network takes the input feature map $U \in \mathbb{R}^{H \times W \times C}$ with width $W$, height $H$ and $C$ channels and outputs $\theta$, the parameters of the transformation $\mathcal{T}_\theta$ to be applied to the feature map: $\theta = f_{\text{loc}}(U)$. The size of $\theta$ can vary depending on the transformation type that is parameterised, *e.g.* for an affine transformation $\theta$ is 6-dimensional as in (1).

The localisation network function $f_{\text{loc}}()$ can take any form, such as a fully-connected network or a convolutional network, but should include a final regression layer to produce the transformation parameters $\theta$.

## 3.2   Parameterised Sampling Grid

To perform a warping of the input feature map, each output pixel is computed by applying a sampling kernel centered at a particular location in the input feature map (this is described fully in the next section). By *pixel* we refer to an element of a generic feature map, not necessarily an image. In general, the output pixels are defined to lie on a regular grid $G = \{G_i\}$ of pixels $G_i = (x_i^t, y_i^t)$, forming an output feature map $V \in \mathbb{R}^{H' \times W' \times C}$, where $H'$ and $W'$ are the height and width of the grid, and $C$ is the number of channels, which is the same in the input and output.

For clarity of exposition, assume for the moment that $\mathcal{T}_\theta$ is a 2D affine transformation $\mathtt{A}_\theta$. We will discuss other transformations below. In this affine case, the pointwise transformation is

$$\begin{pmatrix} x_i^s \\ y_i^s \end{pmatrix} = \mathcal{T}_\theta(G_i) = \mathtt{A}_\theta \begin{pmatrix} x_i^t \\ y_i^t \\ 1 \end{pmatrix} = \begin{bmatrix} \theta_{11} & \theta_{12} & \theta_{13} \\ \theta_{21} & \theta_{22} & \theta_{23} \end{bmatrix} \begin{pmatrix} x_i^t \\ y_i^t \\ 1 \end{pmatrix} \qquad (1)$$

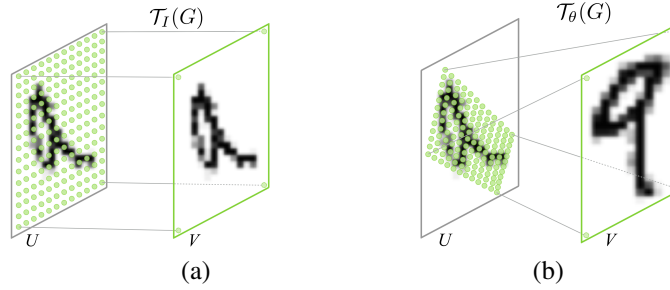

$\mathcal{T}_I(G)$        $\mathcal{T}_\theta(G)$

(a)          (b)

Figure 3: Two examples of applying the parameterised sampling grid to an image $U$ producing the output $V$. (a) The sampling grid is the regular grid $G = \mathcal{T}_I(G)$, where $I$ is the identity transformation parameters. (b) The sampling grid is the result of warping the regular grid with an affine transformation $\mathcal{T}_\theta(G)$.

where $(x_i^t, y_i^t)$ are the target coordinates of the regular grid in the output feature map, $(x_i^s, y_i^s)$ are the source coordinates in the input feature map that define the sample points, and $\mathtt{A}_\theta$ is the affine transformation matrix. We use height and width normalised coordinates, such that $-1 \leq x_i^t, y_i^t \leq 1$ when within the spatial bounds of the output, and $-1 \leq x_i^s, y_i^s \leq 1$ when within the spatial bounds of the input (and similarly for the $y$ coordinates). The source/target transformation and sampling is equivalent to the standard texture mapping and coordinates used in graphics.

The transform defined in (1) allows cropping, translation, rotation, scale, and skew to be applied to the input feature map, and requires only 6 parameters (the 6 elements of $\mathtt{A}_\theta$) to be produced by the localisation network. It allows cropping because if the transformation is a contraction (i.e. the determinant of the left $2 \times 2$ sub-matrix has magnitude less than unity) then the mapped regular grid will lie in a parallelogram of area less than the range of $x_i^s, y_i^s$. The effect of this transformation on the grid compared to the identity transform is shown in Fig. 3.

The class of transformations $\mathcal{T}_\theta$ may be more constrained, such as that used for attention

$$\mathtt{A}_\theta = \begin{bmatrix} s & 0 & t_x \\ 0 & s & t_y \end{bmatrix} \qquad (2)$$

allowing cropping, translation, and isotropic scaling by varying $s$, $t_x$, and $t_y$. The transformation $\mathcal{T}_\theta$ can also be more general, such as a plane projective transformation with 8 parameters, piecewise affine, or a thin plate spline. Indeed, the transformation can have any parameterised form, provided that it is differentiable with respect to the parameters – this crucially allows gradients to be backpropagated through from the sample points $\mathcal{T}_\theta(G_i)$ to the localisation network output $\theta$. If the transformation is parameterised in a structured, low-dimensional way, this reduces the complexity of the task assigned to the localisation network. For instance, a generic class of structured and differentiable transformations, which is a superset of attention, affine, projective, and thin plate spline transformations, is $\mathcal{T}_\theta = M_\theta B$, where $B$ is a target grid representation (*e.g.* in (1), $B$ is the regular grid $G$ in homogeneous coordinates), and $M_\theta$ is a matrix parameterised by $\theta$. In this case it is possible to not only learn how to predict $\theta$ for a sample, but also to learn $B$ for the task at hand.

### 3.3   Differentiable Image Sampling

To perform a spatial transformation of the input feature map, a sampler must take the set of sampling points $\mathcal{T}_\theta(G)$, along with the input feature map $U$ and produce the sampled output feature map $V$.

Each $(x_i^s, y_i^s)$ coordinate in $\mathcal{T}_\theta(G)$ defines the spatial location in the input where a sampling kernel is applied to get the value at a particular pixel in the output $V$. This can be written as

$$V_i^c = \sum_n^H \sum_m^W U_{nm}^c k(x_i^s - m; \Phi_x) k(y_i^s - n; \Phi_y) \;\; \forall i \in [1 \ldots H'W'] \;\; \forall c \in [1 \ldots C] \qquad (3)$$

where $\Phi_x$ and $\Phi_y$ are the parameters of a generic sampling kernel $k()$ which defines the image interpolation (*e.g.* bilinear), $U_{nm}^c$ is the value at location $(n, m)$ in channel $c$ of the input, and $V_i^c$ is the output value for pixel $i$ at location $(x_i^t, y_i^t)$ in channel $c$. Note that the sampling is done identically for each channel of the input, so every channel is transformed in an identical way (this preserves spatial consistency between channels).

In theory, any sampling kernel can be used, as long as (sub-)gradients can be defined with respect to $x_i^s$ and $y_i^s$. For example, using the integer sampling kernel reduces (3) to

$$V_i^c = \sum_n^H \sum_m^W U_{nm}^c \delta(\lfloor x_i^s + 0.5 \rfloor - m)\delta(\lfloor y_i^s + 0.5 \rfloor - n) \qquad (4)$$

where $\lfloor x + 0.5 \rfloor$ rounds $x$ to the nearest integer and $\delta()$ is the Kronecker delta function. This sampling kernel equates to just copying the value at the nearest pixel to $(x_i^s, y_i^s)$ to the output location $(x_i^t, y_i^t)$. Alternatively, a bilinear sampling kernel can be used, giving

$$V_i^c = \sum_n^H \sum_m^W U_{nm}^c \max(0, 1 - |x_i^s - m|)\max(0, 1 - |y_i^s - n|) \qquad (5)$$

To allow backpropagation of the loss through this sampling mechanism we can define the gradients with respect to $U$ and $G$. For bilinear sampling (5) the partial derivatives are

$$\frac{\partial V_i^c}{\partial U_{nm}^c} = \sum_n^H \sum_m^W \max(0, 1 - |x_i^s - m|)\max(0, 1 - |y_i^s - n|) \qquad (6)$$

$$\frac{\partial V_i^c}{\partial x_i^s} = \sum_n^H \sum_m^W U_{nm}^c \max(0, 1 - |y_i^s - n|) \begin{cases} 0 & \text{if } |m - x_i^s| \geq 1 \\ 1 & \text{if } m \geq x_i^s \\ -1 & \text{if } m < x_i^s \end{cases} \qquad (7)$$

and similarly to (7) for $\frac{\partial V_i^c}{\partial y_i^s}$.

This gives us a (sub-)differentiable sampling mechanism, allowing loss gradients to flow back not only to the input feature map (6), but also to the sampling grid coordinates (7), and therefore back to the transformation parameters $\theta$ and localisation network since $\frac{\partial x_i^s}{\partial \theta}$ and $\frac{\partial x_i^s}{\partial \theta}$ can be easily derived from (1) for example. Due to discontinuities in the sampling fuctions, sub-gradients must be used. This sampling mechanism can be implemented very efficiently on GPU, by ignoring the sum over all input locations and instead just looking at the kernel support region for each output pixel.

### 3.4 Spatial Transformer Networks

The combination of the localisation network, grid generator, and sampler form a spatial transformer (Fig. 2). This is a self-contained module which can be dropped into a CNN architecture at any point, and in any number, giving rise to *spatial transformer networks*. This module is computationally very fast and does not impair the training speed, causing very little time overhead when used naively, and even potential speedups in attentive models due to subsequent downsampling that can be applied to the output of the transformer.

Placing spatial transformers within a CNN allows the network to learn how to actively transform the feature maps to help minimise the overall cost function of the network during training. The knowledge of how to transform each training sample is compressed and cached in the weights of the localisation network (and also the weights of the layers previous to a spatial transformer) during training. For some tasks, it may also be useful to feed the output of the localisation network, $\theta$, forward to the rest of the network, as it explicitly encodes the transformation, and hence the pose, of a region or object.

It is also possible to use spatial transformers to downsample or oversample a feature map, as one can define the output dimensions $H'$ and $W'$ to be different to the input dimensions $H$ and $W$. However, with sampling kernels with a fixed, small spatial support (such as the bilinear kernel), downsampling with a spatial transformer can cause aliasing effects.

Finally, it is possible to have multiple spatial transformers in a CNN. Placing multiple spatial transformers at increasing depths of a network allow transformations of increasingly abstract representations, and also gives the localisation networks potentially more informative representations to base the predicted transformation parameters on. One can also use multiple spatial transformers in parallel – this can be useful if there are multiple objects or parts of interest in a feature map that should be focussed on individually. A limitation of this architecture in a purely feed-forward network is that the number of parallel spatial transformers limits the number of objects that the network can model.

| Model | | MNIST Distortion | | | |
|---|---|---|---|---|---|
| | | R | RTS | P | E |
| FCN | | 2.1 | 5.2 | 3.1 | 3.2 |
| CNN | | 1.2 | 0.8 | 1.5 | 1.4 |
| ST-FCN | Aff | 1.2 | 0.8 | 1.5 | 2.7 |
| | Proj | 1.3 | 0.9 | 1.4 | 2.6 |
| | TPS | 1.1 | 0.8 | 1.4 | 2.4 |
| ST-CNN | Aff | 0.7 | 0.5 | 0.8 | 1.2 |
| | Proj | 0.8 | 0.6 | 0.8 | 1.3 |
| | TPS | 0.7 | 0.5 | 0.8 | 1.1 |

Table 1: *Left:* The percentage errors for different models on different distorted MNIST datasets. The different distorted MNIST datasets we test are TC: translated and cluttered, R: rotated, RTS: rotated, translated, and scaled, P: projective distortion, E: elastic distortion. All the models used for each experiment have the same number of parameters, and same base structure for all experiments. *Right:* Some example test images where a spatial transformer network correctly classifies the digit but a CNN fails. (a) The inputs to the networks. (b) The transformations predicted by the spatial transformers, visualised by the grid $T_\theta(G)$. (c) The outputs of the spatial transformers. E and RTS examples use thin plate spline spatial transformers (ST-CNN TPS), while R examples use affine spatial transformers (ST-CNN Aff) with the angles of the affine transformations given. For videos showing animations of these experiments and more see `https://goo.gl/qdEhUu`.

## 4 Experiments

In this section we explore the use of spatial transformer networks on a number of supervised learning tasks. In Sect. 4.1 we begin with experiments on distorted versions of the MNIST handwriting dataset, showing the ability of spatial transformers to improve classification performance through actively transforming the input images. In Sect. 4.2 we test spatial transformer networks on a challenging real-world dataset, Street View House Numbers [21], for number recognition, showing state-of-the-art results using multiple spatial transformers embedded in the convolutional stack of a CNN. Finally, in Sect. 4.3, we investigate the use of multiple parallel spatial transformers for fine-grained classification, showing state-of-the-art performance on CUB-200-2011 birds dataset [28] by automatically discovering object parts and learning to attend to them. Further experiments with MNIST addition and co-localisation can be found in the supplementary material.

### 4.1 Distorted MNIST

In this section we use the MNIST handwriting dataset as a testbed for exploring the range of transformations to which a network can learn invariance to by using a spatial transformer.

We begin with experiments where we train different neural network models to classify MNIST data that has been distorted in various ways: rotation (R); rotation, scale and translation (RTS); projective transformation (P); elastic warping (E) – note that elastic warping is destructive and cannot be inverted in some cases. The full details of the distortions used to generate this data are given in the supplementary material. We train baseline fully-connected (FCN) and convolutional (CNN) neural networks, as well as networks with spatial transformers acting on the input before the classification network (ST-FCN and ST-CNN). The spatial transformer networks all use bilinear sampling, but variants use different transformation functions: an affine transformation (Aff), projective transformation (Proj), and a 16-point thin plate spline transformation (TPS) with a regular grid of control points. The CNN models include two max-pooling layers. All networks have approximately the same number of parameters, are trained with identical optimisation schemes (backpropagation, SGD, scheduled learning rate decrease, with a multinomial cross entropy loss), and all with three weight layers in the classification network.

The results of these experiments are shown in Table 1 (left). Looking at any particular type of distortion of the data, it is clear that a spatial transformer enabled network outperforms its counterpart base network. For the case of rotation, translation, and scale distortion (RTS), the ST-CNN achieves 0.5% and 0.6% depending on the class of transform used for $\mathcal{T}_\theta$, whereas a CNN, with two max-pooling layers to provide spatial invariance, achieves 0.8% error. This is in fact the same error that the ST-FCN achieves, which is without a single convolution or max-pooling layer in its network, showing that using a spatial transformer is an alternative way to achieve spatial invariance. ST-CNN models consistently perform better than ST-FCN models due to max-pooling layers in ST-CNN providing even more spatial invariance, and convolutional layers better modelling local structure. We also test our models in a noisy environment, on $60 \times 60$ images with translated MNIST digits and

| Model | Size | |
| --- | --- | --- |
| | 64px | 128px |
| Maxout CNN [10] | 4.0 | - |
| CNN (ours) | 4.0 | 5.6 |
| DRAM* [1] | 3.9 | 4.5 |
| ST-CNN Single | 3.7 | **3.9** |
| ST-CNN Multi | **3.6** | **3.9** |

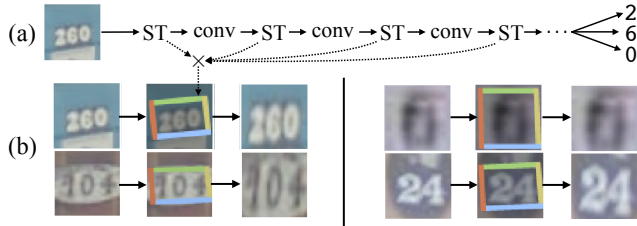

Table 2: *Left:* The sequence error (%) for SVHN multi-digit recognition on crops of $64 \times 64$ pixels (64px), and inflated crops of $128 \times 128$ (128px) which include more background. *The best reported result from [1] uses model averaging and Monte Carlo averaging, whereas the results from other models are from a single forward pass of a single model. *Right:* (a) The schematic of the ST-CNN Multi model. The transformations of each spatial transformer (ST) are applied to the convolutional feature map produced by the previous layer. (b) The result of the composition of the affine transformations predicted by the four spatial transformers in ST-CNN Multi, visualised on the input image.

background clutter (see Fig. 1 third row for an example): an FCN gets 13.2% error, a CNN gets 3.5% error, while an ST-FCN gets 2.0% error and an ST-CNN gets 1.7% error.

Looking at the results between different classes of transformation, the thin plate spline transformation (TPS) is the most powerful, being able to reduce error on elastically deformed digits by reshaping the input into a prototype instance of the digit, reducing the complexity of the task for the classification network, and does not over fit on simpler data *e.g.* R. Interestingly, the transformation of inputs for all ST models leads to a "standard" upright posed digit – this is the mean pose found in the training data. In Table 1 (right), we show the transformations performed for some test cases where a CNN is unable to correctly classify the digit, but a spatial transformer network can.

## 4.2 Street View House Numbers

We now test our spatial transformer networks on a challenging real-world dataset, Street View House Numbers (SVHN) [21]. This dataset contains around 200k real world images of house numbers, with the task to recognise the sequence of numbers in each image. There are between 1 and 5 digits in each image, with a large variability in scale and spatial arrangement.

We follow the experimental setup as in [1, 10], where the data is preprocessed by taking $64 \times 64$ crops around each digit sequence. We also use an additional more loosely $128 \times 128$ cropped dataset as in [1]. We train a baseline character sequence CNN model with 11 hidden layers leading to five independent softmax classifiers, each one predicting the digit at a particular position in the sequence. This is the character sequence model used in [16], where each classifier includes a null-character output to model variable length sequences. This model matches the results obtained in [10].

We extend this baseline CNN to include a spatial transformer immediately following the input (ST-CNN Single), where the localisation network is a four-layer CNN. We also define another extension where before each of the first four convolutional layers of the baseline CNN, we insert a spatial transformer (ST-CNN Multi). In this case, the localisation networks are all two-layer fully connected networks with 32 units per layer. In the ST-CNN Multi model, the spatial transformer before the first convolutional layer acts on the input image as with the previous experiments, however the subsequent spatial transformers deeper in the network act on the convolutional feature maps, predicting a transformation from them and transforming these feature maps (this is visualised in Table 2 (right) (a)). This allows deeper spatial transformers to predict a transformation based on richer features rather than the raw image. All networks are trained from scratch with SGD and dropout [14], with randomly initialised weights, except for the regression layers of spatial transformers which are initialised to predict the identity transform. Affine transformations and bilinear sampling kernels are used for all spatial transformer networks in these experiments.

The results of this experiment are shown in Table 2 (left) – the spatial transformer models obtain state-of-the-art results, reaching 3.6% error on $64 \times 64$ images compared to previous state-of-the-art of 3.9% error. Interestingly on $128 \times 128$ images, while other methods degrade in performance, an ST-CNN achieves 3.9% error while the previous state of the art at 4.5% error is with a recurrent attention model that uses an ensemble of models with Monte Carlo averaging – in contrast the ST-CNN models require only a single forward pass of a single model. This accuracy is achieved due to the fact that the spatial transformers crop and rescale the parts of the feature maps that correspond to the digit, focussing resolution and network capacity only on these areas (see Table 2 (right) (b)

| Model | |
|---|---|
| Cimpoi '15 [4] | 66.7 |
| Zhang '14 [30] | 74.9 |
| Branson '14 [2] | 75.7 |
| Lin '15 [20] | 80.9 |
| Simon '15 [24] | 81.0 |
| CNN (ours)   224px | 82.3 |
| 2×ST-CNN   224px | 83.1 |
| 2×ST-CNN   448px | 83.9 |
| 4×ST-CNN   448px | **84.1** |

Table 3: *Left:* The accuracy (%) on CUB-200-2011 bird classification dataset. Spatial transformer networks with two spatial transformers (2×ST-CNN) and four spatial transformers (4×ST-CNN) in parallel outperform other models. 448px resolution images can be used with the ST-CNN without an increase in computational cost due to downsampling to 224px *after* the transformers. *Right:* The transformation predicted by the spatial transformers of 2×ST-CNN (top row) and 4×ST-CNN (bottom row) on the input image. Notably for the 2×ST-CNN, one of the transformers (shown in red) learns to detect heads, while the other (shown in green) detects the body, and similarly for the 4×ST-CNN.

for some examples). In terms of computation speed, the ST-CNN Multi model is only 6% slower (forward and backward pass) than the CNN.

### 4.3   Fine-Grained Classification

In this section, we use a spatial transformer network with multiple transformers in parallel to perform fine-grained bird classification. We evaluate our models on the CUB-200-2011 birds dataset [28], containing 6k training images and 5.8k test images, covering 200 species of birds. The birds appear at a range of scales and orientations, are not tightly cropped, and require detailed texture and shape analysis to distinguish. In our experiments, we only use image class labels for training.

We consider a strong baseline CNN model – an Inception architecture with batch normalisation [15] pre-trained on ImageNet [22] and fine-tuned on CUB – which by itself achieves state-of-the-art accuracy of 82.3% (previous best result is 81.0% [24]). We then train a spatial transformer network, ST-CNN, which contains 2 or 4 parallel spatial transformers, parameterised for attention and acting on the input image. Discriminative image parts, captured by the transformers, are passed to the part description sub-nets (each of which is also initialised by Inception). The resulting part representations are concatenated and classified with a single softmax layer. The whole architecture is trained on image class labels end-to-end with backpropagation (details in supplementary material).

The results are shown in Table 3 (left). The 4×ST-CNN achieves an accuracy of 84.1%, outperforming the baseline by 1.8%. In the visualisations of the transforms predicted by 2×ST-CNN (Table 3 (right)) one can see interesting behaviour has been learnt: one spatial transformer (red) has learnt to become a head detector, while the other (green) fixates on the central part of the body of a bird. The resulting output from the spatial transformers for the classification network is a somewhat pose-normalised representation of a bird. While previous work such as [2] explicitly define parts of the bird, training separate detectors for these parts with supplied keypoint training data, the ST-CNN is able to discover and learn part detectors in a data-driven manner without any additional supervision. In addition, spatial transformers allows for the use of 448px resolution input images without any impact on performance, as the output of the transformed 448px images are sampled at 224px before being processed.

## 5   Conclusion

In this paper we introduced a new self-contained module for neural networks – the spatial transformer. This module can be dropped into a network and perform explicit spatial transformations of features, opening up new ways for neural networks to model data, and is learnt in an end-to-end fashion, without making any changes to the loss function. While CNNs provide an incredibly strong baseline, we see gains in accuracy using spatial transformers across multiple tasks, resulting in state-of-the-art performance. Furthermore, the regressed transformation parameters from the spatial transformer are available as an output and could be used for subsequent tasks. While we only explore feed-forward networks in this work, early experiments show spatial transformers to be powerful in recurrent models, and useful for tasks requiring the disentangling of object reference frames.

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
