[Supplementary Material]

# Spatial Transformer Networks: Supplementary Material

**Max Jaderberg**     **Karen Simonyan**     **Andrew Zisserman**     **Koray Kavukcuoglu**
Google DeepMind, London, UK
{jaderberg,simonyan,zisserman,korayk}@google.com

## 1   Experiments Continued

In this section we present the results of two further experiments – that of MNIST addition showing spatial transformers acting on multiple objects in Sect. 1.1, and co-localisation in Sect. 1.2 showing the application to semi-supervised scenarios. We also expand upon the details of the experiments presented in the main paper.

### 1.1   MNIST Addition

In this section we demonstrate another use case for multiple spatial transformers in parallel: to model multiple objects. We define an MNIST addition task, where the network must output the sum of the two digits given in the input. Each digit is presented in a separate $42 \times 42$ input channel (giving 2-channel inputs), but each digit is transformed independently, with random rotation, scale, and translation (RTS).

We train fully connected (FCN), convolutional (CNN) and single spatial transformer fully connected (ST-FCN) networks, as well as spatial transformer fully connected networks with two parallel spatial transformers ($2\times$ST-FCN) acting on the input image, each one taking both channels as input and transforming both channels. The two 2-channel outputs of the two spatial transformers are concatenated into a 4-channel feature map for the subsequent FCN. As in Sect. 4.1 of the main paper,

| Model | | RTS |
|---|---|---|
| FCN | | 47.7 |
| CNN | | 14.7 |
| ST-FCN | Aff | 22.6 |
| | Proj | 18.5 |
| | TPS | 19.1 |
| $2\times$ST-FCN | Aff | 9.0 |
| | Proj | 5.9 |
| | TPS | 5.8 |

Table 1: *Left:* The percentage error for the two digit MNIST addition task, where each digit is transformed independently in separate channels, trained by supplying only the label of the sum of the two digits. The use of two spatial transformers in parallel, $2\times$ST-FCN, allows the fully-connected neural network to become invariant to the transformations of each digit, giving the lowest error. All the models used for each column have approximately the same number of parameters. *Right:* A test example showing the learnt behaviour of each spatial transformer (using a thin plate spline (TPS) transformation). The 2-channel input (the blue bar denotes separation between channels) is fed to two independent spatial transformers, ST1 and ST2, each of which operate on both channels. The outputs of ST1 and ST2 and concatenated and used as a 4-channel input to a fully connected network (FCN) which predicts the addition of the two original digits. During training, the two spatial transformers co-adapt to focus on a single channel each.

| Class | MNIST Distortion | |
|---|---|---|
| | T | TC |
| 0 | 100 | 81 |
| 1 | 100 | 82 |
| 2 | 100 | 88 |
| 3 | 100 | 75 |
| 4 | 100 | 94 |
| 5 | 100 | 84 |
| 6 | 100 | 93 |
| 7 | 100 | 85 |
| 8 | 100 | 89 |
| 9 | 100 | 87 |

Table 2: *Left:* The percent of correctly co-localised digits for different MNIST digit classes, for just translated digits (T), and for translated digits with clutter added (TC). *Right:* The optimisation architecture. We use a hinge loss to enforce the distance between the two outputs of the spatial transformer (ST) to be less than the distance to a random crop, hoping to encourage the spatial transformer to localise the common objects.

all networks have the same number of parameters, and are all trained with SGD to minimise the multinomial cross entropy loss for 19 classes (the possible addition results 0-18).

The results are given in Table 1 (left). Due to the complexity of this task, the FCN reaches a minimum error of 47.7%, however a CNN with max-pooling layers is far more accurate with 14.7% error. Adding a single spatial transformer improves the capability of an FCN by focussing on a single region of the input containing both digits, reaching 18.5% error. However, by using two spatial transformers, each transformer can learn to focus on transforming the digit in a single channel (though receiving both channels as input), visualised in Table 1 (right). The transformers co-adapt, producing stable representations of the two digits in two of the four output channels of the spatial transformers. This allows the $2\times$ST-FCN model to achieve 5.8% error, far exceeding that of other models.

## 1.2   Co-localisation

In this experiment, we explore the use of spatial transformers in a semi-supervised scenario – co-localisation. The co-localisation task is as follows: given a set of images that are assumed to contain instances of a common but unknown object class, localise (with a bounding box) the common object. Neither the object class labels, nor the object location ground truth is used for optimisation, only the set of images.

To achieve this, we adopt the supervision that the distance between the image crop corresponding to two correctly localised objects is smaller than to a randomly sampled image crop, in some embedding space. For a dataset $\mathcal{I} = \{I_n\}$ of $N$ images, this translates to a triplet loss, where we minimise the hinge loss

$$\sum_n^N \sum_{m \neq n}^M \max(0, \|e(I_n^\mathcal{T}) - e(I_m^\mathcal{T})\|_2^2 - \|e(I_n^\mathcal{T}) - e(I_n^{\mathrm{rand}})\|_2^2 + \alpha) \qquad (1)$$

where $I_n^\mathcal{T}$ is the image crop of $I_n$ corresponding to the localised object, $I_n^{\mathrm{rand}}$ is a randomly sampled patch from $I_n$, $e()$ is an encoding function and $\alpha$ is a margin. We can use a spatial transformer to act as the localiser, such that $I_n^\mathcal{T} = \mathcal{T}_\theta(I_n)$ where $\theta = f_{\mathrm{loc}}(I_n)$, interpreting the parameters of the transformation $\theta$ as the bounding box of the object. We can minimise this with stochastic gradient descent, randomly sampling image pairs $(n, m)$.

We perform co-localisation on translated (T), and also translated and cluttered (TC) MNIST images. Each image, a $28 \times 28$ pixel MNIST digit, is placed in a uniform random location in a $84 \times 84$ black background image. For the cluttered dataset, we also then add 16 random $6 \times 6$ crops sampled from the original MNIST training dataset, creating distractors. For a particular co-localisation optimisation, we pick a digit class and generate 100 distorted image samples as the dataset for the experiment. We use a margin $\alpha = 1$, and for the encoding function $e()$ we use the CNN trained for digit classification from Sect. 4.1 of the main paper, concatenating the three layers of activations (two hidden layers and the classification layer without softmax) to form a feature descriptor. We

Figure 1: A look at the optimisation dynamics for co-localisation. Here we show the localisation predicted by the spatial transformer for three of the 100 dataset images after the SGD step labelled below. By SGD step 180 the model has process has correctly localised the three digits. An animation of this process can be found in the video in the supplementary material.

use a spatial transformer parameterised for attention (scale and translation) where the localisation network is a 100k parameter CNN consisting of a convolutional layer with eight $9 \times 9$ filters and a 4 pixel stride, followed by $2 \times 2$ max pooling with stride 2 and then two 8-unit fully-connected layers before the final 3-unit fully-connected layer.

The results are shown in Table 2. We measure a digit to be correctly localised if the overlap (area of intersection divided by area of union) between the predicted bounding box and groundtruth bounding box is greater than 0.5. Our co-localisation framework is able to perfectly localise MNIST digits without any clutter with 100% accuracy, and correctly localises between 75-93% of digits when there is clutter in the images. An example of the optimisation process on a subset of the dataset for "8" is shown in Fig. 1. This is surprisingly good performance for what is a simple loss function derived from simple intuition, and hints at potential further applications in tracking problems.

## 1.3 Distorted MNIST Details

In this section we expand upon the details of the distorted MNIST experiments in Sect. 4.1 of the main paper.

**Data.** The rotated dataset (R) was generated from rotating MNIST training digits with a random rotation sampled uniformly between $-90°$ and $+90°$. The rotated, translated, and scaled dataset (RTS) was generated by randomly rotating an MNIST digit by $+45°$ and $-45°$, randomly scaling the digit by a factor of between 0.7 and 1.2, and placing the digit in a random location in a $42 \times 42$ image, all with uniform distributions. The projected dataset (P) was generated by scaling a digit randomly between 0.75 and 1.0, and stretching each corner of an MNIST digit by an amount sampled from a normal distribution with zero mean and 5 pixel standard deviation. The elastically distorted dataset (E) was generated by scaling a digit randomly between 0.75 and 1.0, and then randomly peturbing 16 control points of a thin plate spline arranged in a regular grid on the image by an amount sampled from a normal distribution with zero mean and 1.5 pixel standard deviation. The translated and cluttered dataset (TC) is generated by placing an MNIST digit in a random location in a $60 \times 60$ black canvas, and then inserting six randomly sampled $6 \times 6$ patches of other digit images into random locations in the image.

**Networks.** All networks use rectified linear non-linearities and softmax classifiers. All FCN networks have two hidden fully connected layers followed by a classification layer. All CNN networks have a $9 \times 9$ convolutional layer (stride 1, no padding), a $2 \times 2$ max-pooling layer with stride 2, a subsequent $7 \times 7$ convolutional layer (stride 1, no padding), and another $2 \times 2$ max-pooling layer with stride 2 before the final classfication layer. All spatial transformer (ST) enabled networks place the ST modules at the beginning of the network, and have three hidden layers in their localisation networks with 32 unit fully connected layers for ST-FCN networks and two 20-filter $5 \times 5$ convo-

Figure 2: The architecture of the 2×ST-CNN 448px used for bird classification. A single localisation network $f_{\text{loc}}$ predicts two transformation parameters $\theta_1$ and $\theta_2$, with the subsequent transforms $\mathcal{T}_{\theta_1}$ and $\mathcal{T}_{\theta_2}$ applied to the original input image.

lutional layers (stride 1, no padding) acting on a $2\times$ downsampled input, with $2 \times 2$ max-pooling between convolutional layers, and a 20 unit fully connected layer following the convolutional layers. Spatial transformer networks for TC and RTS datasets have average pooling after the spatial transformer to downsample the output of the transformer by a factor of 2 for the classification network. The exact number of units in FCN and CNN based classification models varies so as to always ensure that all networks for a particular experiment contain the same number of learnable parameters (around 400k). This means that spatial transformer networks generally have less parameters in the classification networks due to the need for parameters in the localisation networks. The FCNs have between 128 and 256 units per layer, and the CNNs have between 32 and 64 filters per layer.

**Training.** All networks were trained with SGD for 150k iterations, the same hyperparameters (256 batch size, 0.01 base learning rate, no weight decay, no dropout), and same learning rate schedule (learning rate reduced by a factor of ten every 50k iterations). We initialise the network weights randomly, except for the final regression layer of localisation networks which are initialised to regress the identity transform (zero weights, identity transform bias). We perform three complete training runs for all models with different random seeds and report average accuracy.

### 1.4 Street View House Numbers Details

For the SVHN experiments in Sect. 4.2 of the main paper, we follow [1, 2] and select hyperparameters from a validation set of 5k images from the training set. All networks are trained for 400k iterations with SGD (128 batch size), using a base learning rate of 0.01 decreased by a factor of ten every 80k iterations, weight decay set to 0.0005, and dropout at 0.5 for all layers except the first convolutional layer and localisation networks. The learning rate for localisation networks of spatial transformer networks was set to a tenth of the base learning rate.

We adopt the notation that conv[$N,w,s,p$] denotes a convolutional layer with $N$ filters of size $w \times w$, with stride $s$ and $p$ pixel padding, fc[$N$] is a fully connected layer with $N$ units, and max[$s$] is a $s \times s$ max-pooling layer with stride $s$. The CNN model is: conv[48,5,1,2]-max[2]-conv[64,5,1,2]-conv[128,5,1,2]-max[2]-conv[160,5,1,2]-conv[192,5,1,2]-max[2]-conv[192,5,1,2]-conv[192,5,1,2]-max[2]-conv[192,5,1,2]-fc[3072]-fc[3072]-fc[3072], with rectified linear units following each weight layer, followed by five parallel fc[11] and softmax layers for classification (similar to that in [4]). The ST-CNN Single has a single spatial transformer (ST) before the first convolutional layer of the CNN model – the ST's localisation network architecture is as follows: conv[32,5,1,2]-max[2]-conv[32,5,1,2]-fc[32]-fc[32]. The ST-CNN Multi has four spatial transformers, one before each of the first four convolutional layers of the CNN model, and each with a simple fc[32]-fc[32] localisation network.

We initialise the network weights randomly, except for the final regression layer of localisation networks which are initialised to regress the identity transform (zero weights, identity transform bias). We performed two full training runs with different random seeds and report the average accuracy obtained by a single model.

### 1.5 Fine Grained Classification Details

In this section we describe our fine-grained image classification architecture from Sect. 4.3 of the main paper in more detail. For this task, we utilise the spatial transformers as a differentiable atten-

tion mechanism, where each transformer is expected to automatically learn to focus on discriminative object parts. Namely, each transformer predicts the location (x,y) of the attention window, while the scale is fixed to $50\%$ of the image size. The transformers sample $224 \times 224$ crops from the input image, each of which is then described each by its own CNN stream, thus forming a multi-stream architecture (shown in Fig. 2). The outputs of the streams are 1024-D crop descriptors, which are concatenated and classified with a 200-way softmax classifier.

As the main building block of our network, we utilise the state-of-the-art Inception architecture with batch normalisation [3], pre-trained on the ImageNet Challenge (ILSVRC) dataset. Our model achieves $27.1\%$ top-1 error on the ILSVRC validation set using a single image crop (we only trained on single-scale images, resized so that the smallest side is 256). The crop description networks employ the Inception architecture with the last layer (1000-way ILSVRC classifier) removed, so that the output is a 1024-D descriptor.

The localisation network is shared across *all* the transformers, and was derived from Inception in the following way. Apart from the ILSVRC classification layer, we also removed the last pooling layer to preserve the spatial information. The output of this truncated Inception net has $7 \times 7$ spatial resolution and 1024 feature channels. On top of it, we added three weight layers to predict the transformations: (i) $1 \times 1$ convolutional layer to reduce the number of feature channels from 1024 to 128; (ii) fully-connected layer with 128-D output; (iii) fully-connected layer with $2N$-D output, where $N$ is the number of transformers (we experimented with $N = 2$ and $N = 4$).

We note that we did not strive to optimise the architecture in terms of the number of parameters and the computation time. Our aim was to investigate whether spatial transformer networks are able to automatically discover meaningful object parts when trained just on image labels, which we confirmed both quantitatively and qualitatively (Sect. 4.3 of the main paper).

We evaluated two input images sizes for the spatial transformers: $224 \times 224$ and $448 \times 448$. In the latter case, we added a fixed $2\times$ downscaling layer before the localisation net, so that its input is still $224 \times 224$. The difference between the two settings lies in the size of the image from which sampling is performed (224 vs 448), with 448 better suited for sampling small-scale crops. The output of the transformers are $224 \times 224$ crops in both cases (so that they are compatible with crop description Inception nets). When training, we utilised conventional augmentation in the form of random sampling ($224 \times 224$ from $256 \times S$ and $448 \times 448$ from $512 \times S$ where $S$ is the largest image side) and horizontal flipping. The localisation net was initialised to tile the image plane with the spatial transformer crops. The evaluation was performed in a conventional manner by considering 5 input samples (four image corners and centre) and 2 horizontal flips, for a total of 10 samples; the predicted softmax posteriors were averaged across these samples to obtain the final class prediction for an image.

We also experimented with more complex transformations (location and scale, as well as affine), but observed similar results. This can be attributed to the very small size of the training set (6k images, 200 classes), and we noticed severe over-fitting in all training scenarios. The hyper-parameters were estimated by cross-validation on the training set.