[Reviews · NeurIPS 2015]

Submitted by Assigned_Reviewer_1

This work addresses the question of how to improve the invariance properties of Convolutional Neural Networks. It introduces the so-called spatial transformer, a layer that performs an adaptive warping of incoming feature maps, thus generalizing the recent attention mechanisms for images. The resulting model requires no extra supervision and is trained back-to-back using backpropagation, leading to state-of-the-art results on several classification tasks.

The paper is clearly written and its main contribution, the spatial transformer layer, is valuable for its novelty, simplicity and effectiveness. The related work section covers most relevant literature, except perhaps recent works that combine deformable parts models with CNNs (see for example "Deformable Part Models are Convolutional Neural Networks", "End-to-End Integration of a Convolution Network, Deformable Parts Model and Non-Maximum Suppression" both at cvpr 2015), since they also incorporate an inference over deformation or registration parameters, as in the spatial transformer case. The numerical experiments cover some relatively challenging datasets (although not large scale tasks such as Imagenet or general object recognition) which highlight the gain produced by the registration layers.

Here are some detailed comments:

--> relationship with a general bilinear model. The spatial transformer applies a linear transform A(x,theta(x)) to the input feature map x, which is itself indexed by a low-dimensional vector of parameters theta(x) that are adapted to the input. This is a particular instance of a general bilinear map, in which a given family of transformations (warpings or affine maps) is considered. Looking at the experiments, it seems as if the transformer is asking for more capacity, since the richer class (thin plate spline) systematically improves over simpler models of transformations. One could then attempt to make the bilinear model a bit more expressive by replacing the class of warping operators by a learnt subfamily of linear operators. Another related comment is whether it would be interesting to consider 3D warpings across feature maps as well. That could perhaps allow the network to model relationships across features with extra capacity.

--> relationship with data augmentation. In order to increase the invariance/stability of the network to large displacements or transformations out of the scope of the pooling layers, one typically resorts to data augmentation. How does the spatial transformer relate to this method? can they work together? One could argue that data augmentation uses prior information, whereas the spatial transformer does not assume classes to be invariant to known transformations.

--> Another comment is whether one could use the learnt registration parameters to construct adversarial examples/data augmentation. In the model, there is a relationship between the so-called localisation parameters theta(x) and x. If say x'= T_beta(x), where T_beta is assumed to belong to the family currently implemented by the spatial transformer,

then it is reasonable to expect to have theta(x') = theta(x)+beta, assuming also a group structure in the space of transformations. In any case, the localisation net should satisfy T_{theta(x)} T_beta = T_{theta( T_beta(x) ) } for all x and beta. Could this property help regularize the localisation net?

--> The paper presents arguments in favor of the spatial transformers together with experiments that show its efficiency. However, the authors do not mention any scenario where the method could fail, or even hurt the performance of vanilla CNNs. When does it fail, or what are its limitations? For instance, aliasing can be potentially dangerous, as pointed out briefly, but also bilinear interpolation is known to be a bad kernel if one needs to preserve high frequency content, which might be important in, say, texture classification or complex object recognition tasks.

Summary: This paper introduces a trainable layer that transforms a feature map following a parametrized diffeomorphism. When used within

Convolutional networks it improves state-of-the-art results on several benchmarks. This work is appealing since it improves the amount of invariance CNN can learn with little extra computational and learning cost.

Despite some remarks relating to its scalability and learning complexity, I recommend this work for publication.

Submitted by Assigned_Reviewer_2

This paper presents a novel layer that can be used in convolutional neural networks. A spatial transformer layer computes re-sampling points of the signal based on another neural network. The suggested transformations include scaling, cropping, rotations and non-rigid deformation whose paramerters are trained end-to-end with the rest of the model. The resulting re-sampling grid is then used to create a new representation of the underlying signal through bi-linear or nearest neighbor interpolation. This has interesting implications: the network can learn to co-locate objects in a set of images that all contain the same object, the transformation parameter localize the attention area explicitly, fine data resolution is restricted to areas important for the task. Furthermore, the model improves over previous state-of-the-art on a number of tasks.

Strength and weaknesses:

+ Interesting approach to jointly solve for certain variation in data.

+ Wide set of experiments that show the performance of the proposed method.

The mathematical derivation is sound, the paper is well written. This work investigates the important problem of encoding invariances in CNN archtectures and therefore is of relevance.

Some questions to the authors:

- It is curious that the latent representation that is learned is so similar to the un-distorted data. Do you have any intutions about this? Is this due to the selected class of transformations? Did you observe other local solutions that resulted in representation that do not have a clear semantic explanation like the ones presented in the paper?

- Another perspective on this is that this extends the idea of a spatial convolution in CNNs. Something similar -- though not learn-able in the re-sampling -- has be presented in "Spectral Networks and Locally Connected Networks on Graphs" (Bruna et al. ICLR 2014), "Sparse Convolutional Networks using the Permutohedral Lattice" (Kiefel et al., arXiv:1503.04949). I would like to see related work like "Modeling Local and Global Deformations in Deep Learning: Epitomic Convolution, Multiple Instance Learning, and Sliding Window Detection" (Papandreou et al., CVPR 2015) be discussed. They also try to tackle the problem of deformations in the input data.

- This work seems to only use the newly defined layer directly on the distorted inputs. What happens when you apply in later layers of the network?

- Do you release the code for training and evaluation?

Summary: This paper extents common convolutional neural network architectures by a spatial transformation layer. This layer re-samples the image/patch according to a parameterized operation like cropping, scaling, rotation,..., etc. The presented work is applied in a wide range of experiments to MNIST, Street View House Numbers, and the CUB-200 birds dataset.

Submitted by Assigned_Reviewer_3

The paper proposes a novel module for CNNs that allows for end-to-end learning of parametric transformations of the inputs (the image or output feature channels from a previous layer). The module has one mini neural network that regresses on the parameters of a parametric transformation, e.g. affine), then there is a module that applies the transformation to a regular grid and a third more or less "reads off" the values in the transformed positions and maps them to a regular grid, hence underforming the image or previous layer. Gradients for backpropagation in a few cases are derived. The results are mostly of the classic deep learning variety, including mnist and svhn, but there is also the fine-grained birds dataset. The networks with spatial transformers seem to lead to improved results in all cases.

Quality:

The paper has quality, both in terms of presentation and content. All seems correct and pretty.

Clarity:

Extremely clear.

Originality:

It is novel as I can tell. The critical new trick seems to be transforming parametrically a sampling grid which makes it possible to learn end-to-end with backpropagation, unlike previous work using reinforcement learning.

Significance:

It is not completely clear that it will be significant. Will it work on complex vision datasets having multiple objects ? That is the question. The birds experiment is the one giving the most hope, but bird parts are mostly a torso and a head. Only more experiments will allow to accurately gauge significance. In any case people should be interested.
Summary: The idea is interesting and backed up with many experiments, the paper is well written and clear, and good.

Author Feedback
Author rebuttal: Thanks to the reviewers for the insightful comments and questions, we will endeavour to incorporate them in the final version.

R1, R3: "recent works...[deformable parts models, epitomic CNNs]"
Thanks for the references, these papers are good to mention as they implement other types of spatial models in a CNN framework, so references will be added.

R1: "attempt to make the bilinear model a bit more expressive by..."
Yes indeed -- we hint at line 199 that this general bilinear form can be completely learned and can be shaped by imposing different regularisation (e.g. low rank). In fact one could attempt to learn a model which regresses the grid directly without any structure imposed on the transformation, however in practice we find this suffers from noisy gradients. Any other differentiable transformation function T_theta could be used instead -- this could be useful if one has some prior on the expected transformations to be incorporated into the spatial transformer network.

R1: "consider 3D warpings across feature maps"
We agree that this is an interesting and natural extension that should be looked into.

R1: "relationship with data augmentation"
A spatial transformer offers a way to achieve equivariance/invariance which is complementary to data augmentation. Instead of modelling invariance with pooling layers and also potentially modelling multiple transformed versions of data with the parameters of the network, a spatial transformer explicitly disentangles the data from its transformation, achieving data invariance and transformation equivariance. But data augmentation is still relevant to STNs, as it extends the training set, helping the STN to learn a better transformation model.

R1: "Could the ground-truth transformation (from data augmentation) help regularize the localisation net?"
Incorporating additional constraints and/or losses on the localisation net (such as the one suggested) would certainly be an interesting direction to explore.

R1: "When does it fail or what are its limitations"
We haven't found any scenarios where it hurts performance, though training stability can be an issue. Preliminary experiments on ImageNet have shown that it is scalable but doesn't affect the accuracy, as the identity transform (or some small perturbation on the identity) is learned (possibly due to ImageNet requiring the use of background context). The bilinear kernel could indeed be a bottleneck, so other kernels should be explored.

R3: "latent representation that is learned is so similar to the un-distorted data"
This is often found to be the case. We comment on this on line 343. This is probably due to the fact that the un-distorted pose is the mean pose of the distorted MNIST datasets. The mean pose is the pose requiring the least extreme transformation across all data samples, so it is the easiest to learn. However, we have observed that due to random ordering of presented training data, sometimes a random bias towards a non-mean pose of training data can cause a bias in the localisation network to transform to a pose that is not the same as the un-distorted data.

R3: "What happens when you apply the STN in the later layers of the network?"
We make use of transformers at later layers in the network in our SVHN experiments (the ST-CNN Multi model, see Table 2 right (a)). It allows us to use much smaller localisation networks as the features, extracted deep in the network, can encode the object pose. Though not presented, we performed similar experiments on MNIST (with the transformer after the first convolutional layer) and this gave very similar results.

R5: "Table 1: could show error rates on MNIST without distortions too"
Thanks, we will look at adding this to the final version.

R5: "fully connected transformer layers... will blow up number of parameters"
Since there are only 32 units in the FC layers of the localisation nets for the ST-CNN Multi model, it is not a huge increase in the number of parameters. The base CNN has 28M params, the ST-CNN Single has 29M params, and the ST-CNN Multi has 31M params. The extra params for ST-CNN Multi are 300k from ST1, 1.1M from ST2, 1.5M from ST3, and 700k from ST4. We will be more explicit on the number of parameters.